# Nickel Nanoparticles: Applications and Antimicrobial Role against Methicillin-Resistant *Staphylococcus aureus* Infections

**DOI:** 10.3390/antibiotics11091208

**Published:** 2022-09-07

**Authors:** Elham Zarenezhad, Hussein T. Abdulabbas, Mahrokh Marzi, Esraa Ghazy, Mohammad Ekrahi, Babak Pezeshki, Abdolmajid Ghasemian, Amira A. Moawad

**Affiliations:** 1Noncommunicable Diseases Research Center, Fasa University of Medical Sciences, Fasa 7461686688, Iran; 2Department of Medical Laboratory Techniques, Faculty of Health and Medical Techniques, Imam Ja’afar Al-Sadiq University, Al Muthanna 9647555434, Iraq; 3Department of pharmacy, Al-Rasheed University College, Baghdad 9643675544, Iraq; 4Friedrich-Loeffler-Institut (Federal Research Institute for Animal Health), Institute of Bacterial Infections and Zoonoses, Naumburger Str. 96a, 07743 Jena, Germany; 5Animal Health Research Institute, Agriculture Research Center (ARC), Giza 12618, Egypt

**Keywords:** methicillin-resistant *Staphylococcus aureus*, nickel nanoparticles, antibacterial effects

## Abstract

Methicillin-resistant *Staphylococcus aureus* (MRSA) has evolved vast antibiotic resistance. These strains contain numerous virulence factors facilitating the development of severe infections. Considering the costs, side effects, and time duration needed for the synthesis of novel drugs, seeking efficient alternative approaches for the eradication of drug-resistant bacterial agents seems to be an unmet requirement. Nickel nanoparticles (NiNPs) have been applied as prognostic and therapeutic cheap agents to various aspects of biomedical sciences. Their antibacterial effects are exerted via the disruption of the cell membrane, the deformation of proteins, and the inhibition of DNA replication. NiNPs proper traits include high-level chemical stability and binding affinity, ferromagnetic properties, ecofriendliness, and cost-effectiveness. They have outlined pleomorphic and cubic structures. The combined application of NiNPs with CuO, ZnO, and CdO has enhanced their anti-MRSA effects. The NiNPs at an approximate size of around 50 nm have exerted efficient anti-MRSA effects, particularly at higher concentrations. NiNPs have conferred higher antibacterial effects against MRSA than other nosocomial bacterial pathogens. The application of green synthesis and low-cost materials such as albumin and chitosan enhance the efficacy of NPs for therapeutic purposes.

## 1. Background

*Staphylococcus* (*S.*) *aureus* is a ubiquitous pathogenic bacterium. Methicillin-resistant *S. aureus* (MRSA) is among the leading causes of nosocomial pathogens which decipher the multidrug-resistance (MDR) phenotype [1,2,3]. MRSA employs various mechanisms to resist drugs, such as cell wall thickening, efflux of compounds, enzymatic destruction, and target variation [4,5,6].

MRSA infections encompass a wide range of manifestations and are a major life-threatening priority worldwide [7,8]. MRSA is resistant to β-lactam antibiotics. These strains carry a modified penicillin-binding protein (PBP) known as PBP-2a (encoded by either *mecA* or *mecC* gene) inherently exhibiting resistance to various β-lactams such as oxacillin, methicillin, and cefoxitin [9]. MRSA is classified based on genotypic diversity and according to various staphylococcal cassette chromosome *mec* (SCC*mec*) types including SCC*mec* (I-XIII) as substantial markers of MRSA epidemiology. The SCC*mec* gene cassette carries the *mecA* or *mecC* genes in addition to other genes associated with aminoglycosides, macrolides, and fluoroquinolones resistance [10]. MRSA is classified into three main categories including healthcare (HA-MRSA), community (CA-MRSA), and livestock-associated (LA-MRSA) strains each with special virulence and resistance patterns [7,10]. HA-MRSA is responsible for nosocomial infections globally and usually carries SCC*mec* types I, II, or III. CA-MRSA has been recorded in patients without or by negligible contact with healthcare settings. These strains usually carry SCC*mec* elements IV, V, and *pvl* genes. The latter is associated with the Panton–Valentine leukocidin. The LA-MRSA isolates are mainly associated with livestock origins and usually carry SCC*mec* IVa and SCC*mec* V elements. Life-threatening infections caused by LA-MRSA have been previously recorded, highlighting the possibility of zoonotic risk [11,12]. In humans, MRSA can cause severe pyogenic infections of the skin and soft tissue, endocarditis, septic arthritis, pneumonia, osteomyelitis, and otitis media [9,13].

## 2. Pathogenicity of MRSA

Although the colonization of *S. aureus* on host surfaces is not harmful, overcoming the host’s innate immunity leads to invasive deep infections. MRSA causes various cutaneous and deep infections such as folliculitis, impetigo, cutaneous abscesses, pyomyositis, necrotizing pneumonia, and fasciitis [14]. HA-MRSA causes implant or surgical site and catheter-associated infections. Bacteremia-related infections include disseminated infections such as descending urinary tract infections, endocarditis, and osteomyelitis. Thereby, the eradication of the bacterium is a concern considering recurrent infections. The bacterium virulence regulation is exerted by a set of global regulatory circuits (two-component systems, TCS, accessory gene regulatory, Agr, and quorum-sensing, QS) which affect gene expression following environmental signals. Additionally, *S. aureus* responds to internal stimuli in the form of QS autoinduced signaling. AgrBDCA regulates RNA effector RNAIII [15,16].

Biofilm formation is an indispensable mechanism of resistance to antimicrobials and for the environment of host responses. The polysaccharide intercellular adhesin (PIA) mediates the bacterial binding to host cells. Some other major surface adhesin proteins include fibrinogen binding protein (FnBP) A and B, surface binding protein A (Spa), cell wall-anchored proteins (CWP), clumping factors (Clfs) A and B, and surface binding protein (SasG) [17,18,19,20].

Following the bacterial attachment to the cells and colonization, pathogenesis is initiated and developed via the production of toxins, exoenzymes such as exfoliative toxins (ETs), Panton–Valentine leukocidin (PVL), toxic shock syndrome toxin1 (TSST1), phenol-soluble modulins, leukotoxin and haemolysin, lipases, proteases and nucleases, and immunomodulators or immune evasion gene clusters (IEC1 and IEC2) [21].

Phenol-soluble modulins, leukotoxin, and haemolysin are known as pore-forming toxins which lyse the host cells. α-, β-, γ-, and δ-hemolysins of *S. aureus* cause the lysis of erythrocytes, epithelial and endothelial cells, monocytes, damage of the epithelium, and induction of apoptosis. Leukotoxins or PVL target and destroy white blood cells such as macrophages, monocytes, and neutrophils. These include LukDE, LukAB, LukS-PV, and LukF-PV. PVL is associated with soft tissue and skin infections in both MSSA and MRSA [22,23].

Phenol-soluble modulins (PSMs) including PSMα1–PSMα4 play a substantial role in bacterial pathogenesis via cell lysis, inflammation, immune regulation, and biofilm formation or detachment [24]. Exfoliative toxins (ETA-ETD) cause staphylococcal scaled skin syndrome (SSSS) which is associated with dehydration, loss of superficial skin layers, and secondary infections which are not significantly different from MSSA and MRSA [25]. Staphylococcal enterotoxins (SEs) and TSST-1 act as superantigens which are T-cells mitogens. SEs cause food poisoning and gastrointestinal problems such as emesis. TSST1 causes the release of extraordinary amounts of pro-inflammatory cytokines [26]. In addition to humans, MRSA has long been recognized globally to colonize numerous wild and domesticated livestock animals and develop infections [27,28,29,30,31]. The widespread distribution of MRSA among livestock is largely due to the indiscriminate prescription/consumption of antimicrobials for animal breeding or agricultural activities. For example, it has impacted more than 40% of pig farms, 20% of cattle farms, and 20% to 90% of turkey farms in Germany. Numerous studies have shown that there is a high risk of MRSA colonization and infection in humans who come into contact with livestock [32,33]. MRSA is repeatedly recorded in dairy farms as a cause of mastitis with failure in elimination due to its resistance against β-lactam antibiotics employed for related infections [34].

MRSA strains employ various virulence factors to invade the host and develop resistance [6]. The development of resistance to last-resort antimicrobial treatments such as glycopeptides (vancomycin and teicoplanin) is a crisis in the eradication of vancomycin-intermediate *S. aureus* (VISA) or vancomycin-resistant *S. aureus* (VRSA) [17,35]. For skin infections (impetigo) of MRSA, mupirocin and fusidic acid or alternative 1% hydrogen peroxide (H_2_O_2_) are recommended. For abscesses, in conditions of neutropenia, cell-mediated immunity deficiency, or severe infection, there is a need for co-trimoxazole or clindamycin treatment. For cellulitis and soft tissue infections, glycopeptides, tigecycline, and linezolid are useful. Susceptibility testing is strongly recommended prior to the utilization of antimicrobial drugs. Antimicrobial therapy necessity should be evaluated when infections are observed due to costs and risk of resistance evolution. The term “superbug” is defined as a strain developing vast antibiotic resistance [36].

Combination therapies have demonstrated substantial bactericidal effects against MRSA. The combination of carbapenems with linezolid, and each of imipenem/fosfomycin/gentamicin/oxacillin/rifampin and daptomycin have deciphered acceptable anti-MRSA effects [37,38,39]. A β-lactams and daptomycin combination facilitates its binding and mitigates resistance development [40,41]. Additionally, the combination of gentamicin/β-lactams/rifampin and vancomycin has outlined significant effects [42,43]. However, the combination of vancomycin and β-lactams has had nephrotoxicity according to clinical trials [44,45,46].

Natural combination therapies using Polysporin and Neosporin peptides have also exerted anti-MRSA effects [47]. Major chemotherapy approaches to combat MRSA have included telavancin, teicoplanin, ceftaroline, vancomycin, and oxazolidinones [48,49]. The combination of β-lactam and either arbekacin or vancomycin is recommended against MRSA. Notably, granulocyte counts are also recommended for antibiotics consumption [50,51]. Ceftaroline fosamil has exhibited substantial anti-MRSA effects [52,53,54]. Daptomycin and ceftaroline have exhibited higher bactericidal effects and linezolid has exerted strong effects on bacteremia.

As the first clinically applied anti-MRSA oxazolidinones, linezolid is a promising antibiotic with low resistance development to MRSA. However, epidemiological data have unraveled that linezolid resistance among MRSA isolates is in enhancement alongside novel antimicrobials quinupristin/dalfopristin (Q/D), daptomycin, and tigecycline [55,56]. Major drawbacks in the conventional approaches to MRSA eradication using antibiotics include non-specific effects which kill human beneficial bacteria, adverse/toxic effects, drug resistance development, and the high cost and time-consuming development of new drugs.

## 3. Nanoparticle Applications to Combat MRSA

The efficient, non-toxic, proper, accurate, and cost-effective eradication of MRSA infections has been recently achieved through the application of nanocarriers or nanoparticles (NPs) [57,58]. NPs have a size range of 1 to 100 nm. The physicochemical properties of NPs and lower costs are gaining attention for use as antimicrobial agents [59,60,61,62]. NPs have exhibited antimicrobial effects, particularly those synthesized using green methods [6]. NPs can also be used for the detection of MRSA [63]. The main mechanisms of NP antibacterial effects include an impairment in metabolism or bacterial integrity (CuNPs), replication and transcription disruption, tRNA, ATPases, membrane-bound enzymes and biofilm inhibition, protein denaturation (AgNPs), and reactive oxygen species (ROS) production [64,65,66]. Various NPs such as silver (Ag), gold (Au), and lower-cost NPs such as nickel (Ni), titanium-oxide, (TiO), zinc oxide (ZnO), silica (SiO2), and bismuth (Bi) NPs have deciphered efficient bactericidal effects against MRSA in vitro and in vivo [67,68,69,70,71]. Various methods of NP delivery to cells for antibacterial activity include polymeric NPs, liposomes, carbon NPs, and metal or metal-oxide NPs. ZnO could eliminate MRSA skin infection at 1875 mg/mL possibly via amino acid synthesis inhibition [70]. TiO2 NPs produce free radicals which kill MRSA [72]. Cu-doped ZnO nanorods have exhibited more potential effects than that of ZnO singly [73]. Cefotax-based magnetic NPs have exhibited promising anti-MRSA effects against isolates originating from livestock and dairy sources [58]. Interestingly, S,N-GQDs/NiO nanocomposites have exerted extraordinary anti-*S. aureus* effects in vitro with minimum inhibitory (MIC) and bactericidal (MBC) concentrations values of 0.4 and 0.8 mg/mL, respectively [74]. Nickel oxide nanoparticles (NiO NPs) exhibit promising traits such as biocompatibility, thermal and chemical stability, and interesting optical characteristics. The development of NiO nanocomposites has improved their bactericidal effects. The simple synthesis of a CdO–NiO–ZnO nanocomposite using the microwave method also exhibited antibacterial effects against Gram-positive and Gram-negative species [75,76]. In addition, NiOCuO-10%RGO also demonstrated substantial antibacterial activity [74]. NPs can specifically carry drugs and enter into cells via endocytosis. A few NPs for the eradication of MRSA have been used for clinical trials. PLGA-rifampicin NPs have decreased the MIC against MRSA from 0.0008 to 0.002 µg/mL [77]. Vancomycin-loaded hydroxyapatite increased the vancomycin release and improved bone regeneration [78]. The combination of levofloxacin and AgNPs unraveled a synergistic effect with 0.5 and 10 µM against control strains and MRSA, respectively [79]. The application of liposome-albumin-vancomycin has decreased toxicity and improved the anti-MRSA activity [80]. The liposomal formulation of amphotericin-b (AmBisome^®^) has also enhanced anti-MRSA efficiency and mitigated nephrotoxicity [81]. Various NP-based drug delivery systems used for skin treatment have included solid lipid nanoparticles, nanostructured lipid carriers, liposomes, niosomes, and nanoemulsions [82].

It was revealed that Chitosan/Gold Nanoparticle/Graphene Oxide could separate, identify, and eradicate MRSA superbugs within water contaminants. Chitosan has a positive charge and can trap these strains. In a study by Gupta, engineered polymer NPs exerted anti-biofilm effects against MRSA strains at non-toxic levels [83].

Mesoporous silica nanoparticles (MSNs) carrying enzymes have exhibited anti-biofilm (dispersal) effects against *S. aureus*, decreasing the bacterial cells efficiently after 24 and 48 h [84], respectively.

## 4. Advantages and Disadvantages of Nanoparticles

The reduced size of particles provides higher surface accessibility for ligand binding, easy and size adoptive production, rapid penetration and accumulation into cells, absorption and carriage of various compounds, higher drug deposition rates, and a lower rate of clearance from the body [85,86]. The facilitated and rapid penetration and long-lasting deposition of NP drugs are promising in regard to the control of MRSA infections. However, permeation enhancers may disrupt the lipids of the stratum corneum. Further challenges are considerable for various formulations of NPs such as drug expelling, encapsulation, physical stability, controlled release, transdermal delivery challenges (lipid carriers), viscosity differences, skin penetration, costs and large-scale production (niosomes), agglomeration, more frequent doses, systemic adsorption (nanocrystals), and risk of toxicity (polymeric NPs) [87,88]. It is worth mentioning that interactions of NPs with the human body enzymes and cytosol may affect host cells, as demonstrated for CuNPs compared to NiNPs, which necessitates the further evaluation of corresponding effects [89].

This review assessed published data between 2015 and 2022 discussing the involvement of NiNP or NiO NP applications in the treatment of MRSA infections. Found publications were screened based on their titles, abstracts, and full-text availability. English language texts were included herein. The flowchart of publications selection and content of this review is illustrated in Figure 1.

## 5. Nanoparticle Features and Synthesis

The diverse applications of NPs in medicine, environment, agriculture, catalysis, biosensing, and cancer theranostics have attracted interest for their synthesis using chemical, physical, and biological methods. NPs are synthesized through various approaches according to corresponding features of morphology, size, stability, and biocompatibility. These are commonly synthesized by microemulsion, hydrothermal, coprecipitation, sol-gel, ball milling, and biological methods [90]. In the physical method of synthesis, NP size and morphology are difficult to adjust, whereas, in chemical methods, it is possible by alterations in reaction conditions [91]. A hydrothermal method is the most suitable chemical method for NP synthesis owing to its easy and versatile nature and controllable synthesis of high crystalline and homogenous particles [92]. Noticeably, chemical methods leave adverse health and environmental effects and also consume large energy. Considering these, green synthesis (microbial and plant extracts) is promising in terms of considerably lower cost, pollution and toxicity, and health and environmental benefits [93]. In spite of seasonal and geographical limitations of herbs, low purity and amounts, green synthesis is preferred for the chemical synthesis of NPs [94].

## 6. Importance of Nickel and Nickel-Oxide Nanoparticles

Nickel nanoparticles (NiNPs) including magnetic metal intermediate-cost particles have been studied vastly thanks to their myriad applications such as for magnetic sensors [95], memory devices [96], and for biomolecular separation [97]. The eventual proficiency of each material or NP reflects and depends on the structure, shape, size, and purity of NiNPs or derived materials. Nickel oxide (NiO) is an inevitably crucial part of today’s nanotechnology and intermediate-cost metal oxide is a cubical lattice structure [98]. The extraordinary chemical stability, high binding affinity, and ferromagnetic properties of NiNPs provide an indispensable field of study which includes their synthesis and application. They are applied mostly because of cost-effectiveness catalysts considering vast natural resource existence and driving reactions by alternate routes. These NPs contain numerous biomedical usages, including cell isolation, medicine delivery, magnetic resonance imaging, biomedical diagnostics, and more [99]. Considerable antibacterial attributes have been reported against *Bacillus subtilis*, *Pseudomonas aeruginosa*, *Klebsiella pneumoniae*, *S. aureus*, and many other microorganisms [100] (Figure 2). In addition, various forms of NP delivery have been represented in Figure 3.

## 7. Mechanism of Action of Nickel Nanoparticles

Antibacterial activities of NiNPs are related to nickel ion content which interpenetrate the bacterial cell and reach the surface of the bacterial cell membrane and intracellular milieu. This influx of nickel cations destroys organelles like ribosomes and affects bacterial metabolism. A study of the literature unravels how all of this happens, due to electrostatic attraction and the use of negatively charged intercellular microbial cells and positively charged nickel ions [103,104]. Nanoparticles exhibit great surface activity due to the large surface-to-volume proportion. Exposure of *E. coli* to NiNPs disrupts membrane morphology and transport. Nickel’s high affinity to sulfur- and phosphor-containing components such as DNA and proteins disrupts DNA replication and causes protein deformation [105].

## 8. Recent Data Regarding NiONP Effects against MRSA

Haghshenas, Leila, et al. [106] appraised the antibacterial efficacy of gold (AuNPs) and NiNPs separately and in combination with *E. coli* and *S. aurous* in milk. Transmission electron microscopy (TEM) and Uv-vis spectroscopy were used to measure the size and shape of these NPs. A broth microdilution procedure was used to assess the minimum inhibitory concentration (MIC) of the NPs. Then, the NPs’ effect on milk was investigated individually and in combination at 25 °C and 50 °C, respectively. These results displayed that AuNPs and NiNPs with mean sizes of 10 nm and 50 nm had a favorable bactericidal effect (*p* ≤ 0.05) against *E. coli* and *S. aureus*. MICs of AuNPs and NiNPs against *S. aurous* included 0.42 and 0.21 μg mL^−1^ and 0.84 and 0.42 μg mL^−1^ for *E. coli*, respectively. The data of this study revealed that the antibacterial effect of NPs in milk is temperature- and dose-dependent, and the greatest degree of effect was observed in the combined concentration at 50 °C. NiNPs were found to have a stronger antibacterial effect than AuNPs. Accordingly, *S. aureus* was more sensitive to NPs than *E. coli*. AuNPs and NiNPs were proposed as effective candidates for combating pathogens in the food system. This may be due to the charge of the cell wall surface which has a higher affinity for NiNPs.

Haider, Ali, et al. [107] developed a green plant (ginger and garlic) to replace the bactericidal and synthetic catalyst in the textile industry, reducing NiO NPs. The synthesis of NPs was corroborated using X-ray diffraction (XRD) and ultra-violet visible spectroscopy (UV-Vis), which has a potent absorption at 350 nm with a size between 16 to 52 nm for ginger and 11 to 59 nm for garlic. Scanning and transmission electron microscopy (SEM and TEM, respectively) confirmed pleomorphism with cubic and spherical NPs. In addition, the exact amounts of garlic and ginger extract (1:3.6 mL) combined for the synthesis of NiO NPs were effectively confirmed using Fourier transform infrared spectroscopy (FTIR). Garlic-reduced phytochemical NPs effectively degraded bactericidal activity against MRSA at higher concentrations (0.5, 1.0 mg/50 µL) as well as methylene blue (MB) dye. Finally, green synthesized NiO NPs were eminent factors for resolving drug resistance as well as eco-friendly catalytic factors that may be selected on an industrial measure.

In a study, NiO NPs were synthesized using photolysis and their anti-biofilm property was evaluated. XRD studies showed the presence of NiO NPs with high crystallinity. NiO particle size ranged from 13 to 31 nm. There were 42 samples of medical waste from various hospitals in Baghdad from 2 October to 12 October 2020. Isolation outcomes were recorded from 15 isolates of *S. aureus*. The results of both methods (well propagation method and PCR) showed that the percentages of MRSA in these two methods were 53.3% and 73.4%, respectively. The outcomes of MRSA biofilm formation showed that only four isolates (36.3%) were not able to produce biofilm using the plate microtiter method. On the other hand, other isolates (63.7%) were able to produce biofilms. Antimicrobial activity of various concentrations (10–100 µg/mL) ranged from 0–13 mm inhibition zones. The MIC concentration was 265 µg/mL (63.7%) from seven MRSA isolates and 530 µg/mL (36.4%) from four isolates. The results outlined that the hemolytic activity of 2.38%, 2.23%, 2.41%, and 2.69% corresponded to 0, 0.01, 0.1 and 1 mg/mL of NiO NPs, respectively [108].

In another study, NiO with sulfur, and nitrogen co-doped-graphene quantum dots-decorated NiO nanocomposites (S, N-GQDs/NiO) were synthesized using an easy hydrothermal technique. Then, their antibacterial activity against *E. coli*, *S. aureus*, *P. aeruginosa*, and MRSA was assessed. The prepared S,N-GQDs/NiO nanocomposites conferred the highest antibacterial effect against *S. aureus* among these species. The S, N-GQDs/NiO nanocomposite NPs exerted the highest effect against *S. aureus* (17 mm) in the disk diffusion method. The attendance of graphene quantum spots in S,N-GQDs/NiO nanocomposites comforts the ROS mechanism that leads to antibacterial activity [74].

A study inferred the structural and morphological properties of NiO NPs synthesized by a novel, convenient, eco-friendly, and cost-effective ash-supported approach [109]. The antibacterial property of NiNPs was attended using pathogenic strains including Gram-negative and Gram-positive bacteria by the cup and well diffusion method. Raman analysis confirmed the formation of pure NiO cubic phase at an annealing temperature of 700 °C. The SEM illuminated the particle morphology and size of 40 to 90 nm. The antibacterial activity of the NiO NPs was studied using three Gram-negative bacteria (*Salmonella typhi*, *Pseudomonas aeruginosa*, *E. coli*) and one Gram-positive bacterium (MRSA). Accordingly, MRSA was sensitive to NiONPs with a diameter of 26 mm inhibition zone [110].

NiO NPs with a green approach were synthesized using *Eucalyptus globulus* leaf extract and their bactericidal traits were measured. The synthesized NiO NPs were pleomorphic and ranged in size from 10 to 20 nm. The X-ray diffraction (XRD) analysis outlined an average size of NiO NPs as 19 nm. The bioactivity experiment showed the antibacterial and anti-biofilm properties of NiO NPs against extended spectrum beta lactamase (ESβL (+))-producing *E. coli*, *P. aeruginosa*, and *S. aureus*. The growth inhibition method showed a time-dependent decrease in the concentration of NiO NPs in the survival of treated cells. The inhibition of NiO NPs-induced biofilms was demonstrated using a sharp increase in the red fluorescence characteristic of propidium iodide (PI) when SEM illustrations of NiO NPs-treated cells were reduced and distorted by obvious depressions/indentations. In general, *E. globulus* leaf extract can be safely used to synthesize eco-friendly NiO NPs with the potential for the elimination of infections affecting human health (Table 1) [111].

In a study, Superparamagnetic Ni@2D-MoS2 nanosheets exerted anti-biofilm and antibacterial effects against MRSA and *E. faecalis* demonstrating the selective removal of these infections [112].

## 9. Future Prospects

Metal NPs such as Au, Ag, Zn, and Cu are believed to be toxic to eukaryotic cells at high levels. Hence proper and target-specific delivery is crucial. Since the stability of metal or metal oxide NPs is not ensured and even PEGylated NPs are removed from the bloodstream, their use for topical infections is more promising. Low-toxic or non-toxic materials such as albumin and chitosan are considerable for the carriage of drugs [113,114]. Considering the low costs of NiNPs, more investigations are needed to assess their efficiency in vivo and in clinical trials. In addition, combination therapies using NiNPs can be helpful for the eradication of extreme drug-resistant strains. As metal NPs have exerted acceptable anti-MRSA effects, the interactions of NPs with human body enzymes and cell structures need to be evaluated by further experiments to uncover their possible detrimental effects on the host [115].

## 10. Conclusions

Considering the drawbacks in MRSA infection eradication due to vast antibiotic resistance and the expression of numerous virulence factors to develop severe infections, particularly among vulnerable individuals, novel treatment choices are warranted. In addition, MRSA develops resistance to last-resort antibiotics during a time span, owing to a non-adherence to the prescription and consumption of antibiotics. Due to the high costs, side effects, and time duration needed for the synthesis of synthesized drugs, seeking efficient alternative approaches for the eradication of drug-resistant bacterial agents seems to be an unmet requirement. NPs have been applied as prognostic and therapeutic agents in various aspects of biomedical sciences. Lower-cost NPs such as NiNPs and their formulations have been utilized as antibacterial agents against MRSA infections. The application of green synthesis and low-cost materials such as albumin and chitosan enhance the efficacy of NPs for therapeutic purposes. NiNPs’ precise mechanism of action can be predicted and understood via in silico, in vitro, and in vivo studies.

## Figures and Tables

**Figure 1 antibiotics-11-01208-f001:**
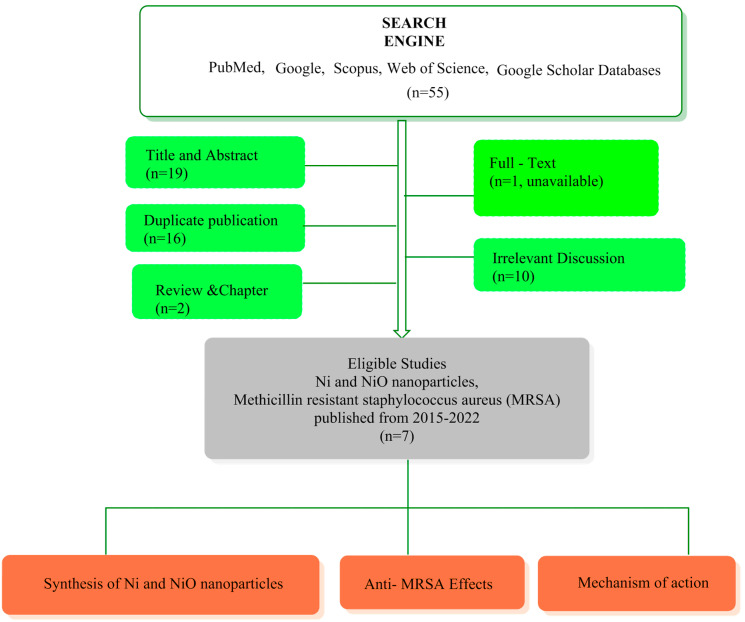
Flowchart of study procedure in this review.

**Figure 2 antibiotics-11-01208-f002:**
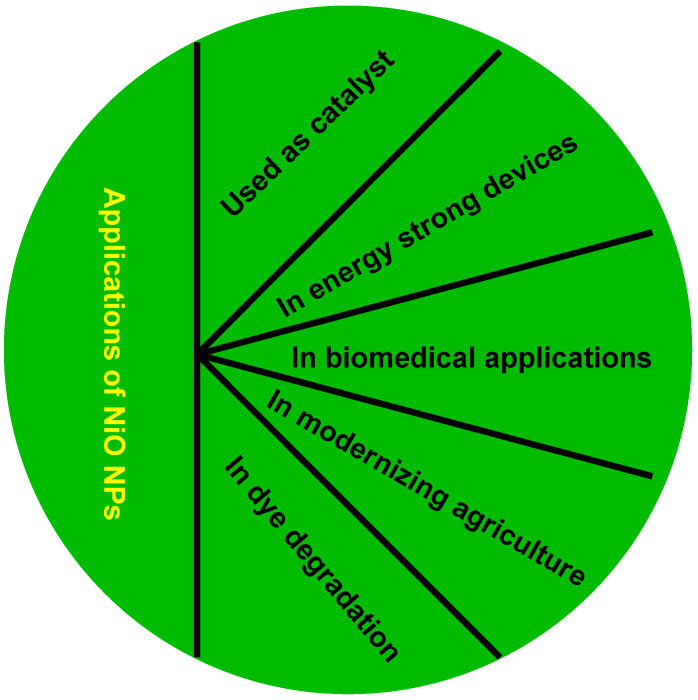
Applications of NiO NPs [92,101,102].

**Figure 3 antibiotics-11-01208-f003:**
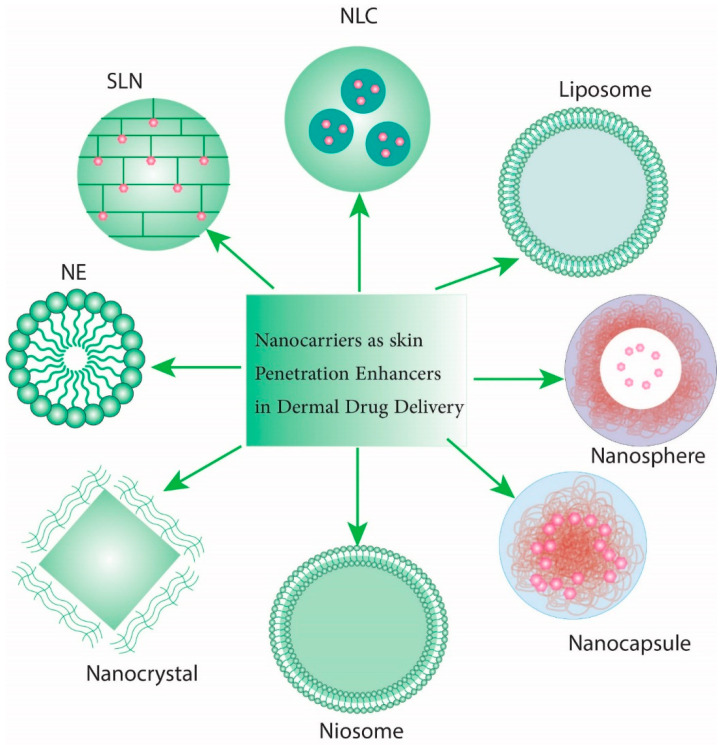
Nanocarriers as skin penetration enhancers in dermal drug delivery. NE: nanoemulsion, SLN: solid lipid nanoparticles, NLC: nanostructured lipid carriers.

**Table 1 antibiotics-11-01208-t001:** NiNP concentrations and conditions against MRSA in vitro.

Nanoparticle	MIC	Mechanism of Action	Conditions	Reference
NiNPs	0.21 (µg/mL)	ND	In vitro	[106]
NiO NPs	1 mg/50 µL	ND	In vitro	[107]
NiO NPs	265 µg/mL	ND	In vitro	[108]
(S,N-GQDs/NiO) NPs	ND *	ND	In vitro	[74]
NiO NPs	ND **	ND	In vitro	[110]
NiO NPs	0.8 mg/mL ***	ND	In vitro	[111]

MIC: minimum inhibitory concentration, ND: not determined, ND *: disk diffusion test (17 mm), ND **: disk diffusion test (17 mm), *** MBC: (minimum bactericidal concentration) included 1.6 mg/mL.

## Data Availability

No new data were created or analyzed in this study. Data sharing is not applicable to this article.

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
