# Peer review of "Nickel Nanoparticles: Applications and Antimicrobial Role against Methicillin-Resistant *Staphylococcus aureus* Infections"

_antibiotics, 2022, doi:10.3390/antibiotics11091208_

Round 1

Author Response

Dear reviewers/editor

Thanks for valuable sent comments which improved this study. The authors have attempted to address queries and improve the manuscript.

Reviewer1

In the present article, the authors have reported Applications of Nickel nanoparticles for the elimination of Methicillin-resistant Staphylococcus aureus Infections. The article is quite informative and is a good piece of work and found significant findings within the scope of this journal. Overall, this is a clear, concise, and well-written manuscript. It will be helpful for the readers. I would suggest some points before publication.

Minor points:

  1. Author should mentioned a few about general idea of nanoparticles and their synthesis approach. Such as chemical, physical and biogenic in 5-7 lines before the section of Importance of Nickel and Nickel Oxide Nanoparticles.

Response:

Dear reviewer, thanks for your valuable comments. The synthesis of NPs by various common methods was added as a section before the section of Importance of Nickel and Nickel Oxide Nanoparticles.

  1. Authors must be cited the recent references which is related similar kind of NPs and their functionalities. To improve of the literature review in introduction section, the authors are recommended to read and add following references to reference part: https://doi.org/10.3390/surfaces5010003 https://doi.org/10.1016/j.bbrep.2022.101320 3. The authors are supposed to review the manuscript once more for major and minor mistakes if available.

Response:

Related papers and several other studies were included in this review.

These sentences were added and cited: “NPs have exhibited antimicrobial effects particularly those synthesized using green methods”. “Considering these, green synthesis (microbial and plant extracts) is promising in terms of considerably lower cost, pollution and toxicity, and health and environmental benefits”.

Author Response

Reviewer 2

  1. Abstract of the reviewer must be well elaborated with key finding made after the reviewer literature. Reviewer article is not just the survey of the published articles but critical listing the progress, gaps identified after in depth comparison of the work. This is lacking in this reviewer work.

Response:

Dear reviewer, thanks for your valuable comments. The abstract was revised after checking main findings of the whole manuscript.

  1. Such statements shouldn’t be written in abstract “However, more in-depth investigations are needed to verify their efficiencies 24 in vivo and in clinical trials”. This reflects and confuse as it is a research article.

Response:

Those confusing and un-related sentences were removed.

  1. Authors mentioned the literature from 2016-2021, but it is observed that literature from other years is also included.

Response:

The papers were checked again. Those related papers during 2015-2022 were included.

  1. Other nanoparticles like silver (Ag), gold (Au), and lower cost NPs such as nickel (Ni), titanium-oxide, (TiO), zinc oxide (ZnO), silica (SiO2) and bismuth (Bi) are also being discussed.

Response:

Various NPs were reviewed in a separate section “Nanoparticles applications to combat MRSA”. However, in the special section “Recent data regarding the NiONPs effects against MRSA”, the mechanism of action of nickel NPs against MRSA has been reviewed.

  1. Some detailed mechanism can be drawn based on literature.

Response:

The mechanisms of action of nickel NPs against MRSA has not been determined in searched studies. However, possible mechanisms were added in the text.

  1. The figures 2 and 3 cited have no relevance to the topic of the review.

Response:

The un-related references were removed.

  1. The Table 1 given in not very comprehensive.

Response:

Table 1 was improved and more details were included.

  1. At least a graph should be drawn to show the number of papers searched out from search engines using keyword.

Response:

The graph of papers was modified.

  1. However, overall paper English and writing style comply with the quality. Just a few miner mistakes are noticed.

Response:

Thanks for your comments. We also checked the whole manuscript.

Reviewer 3 Report

The authors represented interesting research based on "Nickel nanoparticles applications for the elimination of methicillin-resistant Staphylococcus aureus infections". The research topic was novel but need to be grammatically correct. The introduction part is well justified and the authors also formulated correct conclusions. However, I suggest several points to improve the quality of the manuscript.

1.     I suggest authors to revise the title as "Nickel nanoparticles: their applications and role in elimination of methicillin-resistant Staphylococcus aureus infections" of the manuscript to remove grammatical errors

2.     The Authors are suggested to add some references in Point 4 Advantages and Disadvantages of Nanoparticles to properly justify the written statement.

3.     The style of writing and grammar should be improved. There are a lot of grammatical mistakes in the headings of the topics included in the manuscript.

4.     The authors should add a reference in Line 79, of the manuscript.

5.     I suggest authors to revise the heading 5, Methods of the manuscript (Methods for what?). The heading should be written as per the matter it includes.

Author Response

Reviewer3

The authors represented interesting research based on "Nickel nanoparticles applications for the elimination of methicillin-resistant Staphylococcus aureus infections". The research topic was novel but need to be grammatically correct. The introduction part is well justified and the authors also formulated correct conclusions. However, I suggest several points to improve the quality of the manuscript.

  1. I suggest authors to revise the title as "Nickel nanoparticles: their applications and role in elimination of methicillin-resistant Staphylococcus aureus infections" of the manuscript to remove grammatical errors

Response:

Dear reviewer, thanks for your valuable comments. The title was revised.

  1. The Authors are suggested to add some references in Point 4Advantages and Disadvantages of Nanoparticles to properly justify the written statement.

Response:

The references were added.

  1. The style of writing and grammar should be improved. There are a lot of grammatical mistakes in the headings of the topics included in the manuscript.

Response:

The writing and grammar of the manuscript was improved.

  1. The authors should add a reference in Line 79, of the manuscript.

 Response:

The reference was checked.

  1. I suggest authors to revise the heading 5, Methods of the manuscript (Methods for what?). The heading should be written as per the matter it includes.

Response:

The heading was removed and the contents were merged into the text.

Reviewer 4 Report

1.      Line 59… Reference [6,10] [11] should be corrected as [6,10,11]

2.      MRSA strains employ various virulence factors to invade the host and develop resistance….. Need reference Line 79.

3.      Recently, some good reviews are published regarding the application of nanoparticles to combat MRSA e.g. Application of Nanomaterials in the Prevention, Detection, and Treatment of Methicillin-Resistant Staphylococcus aureus (MRSA). Pharmaceutics 2022, 14, 805. https://doi.org/10.3390/ pharmaceutics14040805…  So, what are the advantages of author work over the published one.

Author Response

Reviewer 4

  1. Line 59… Reference [6,10] [11] should be corrected as [6,10,11]

Response:

We thank the reviewer for his/her valuable suggestions to improve the manuscript. The references were corrected.

  1. 2. MRSA strains employ various virulence factors to invade the host and develop resistance….. Need reference Line 79.

Response:

The references were added.

  1. Recently, some good reviews are published regarding the application of nanoparticles to combat MRSA e.g. Application of Nanomaterials in the Prevention, Detection, and Treatment of Methicillin-Resistant Staphylococcus aureus (MRSA). Pharmaceutics 2022, 14, 805. https://doi.org/10.3390/ pharmaceutics14040805… So, what are the advantages of author work over the published one.

Response:

We studied this published paper and used it in our manuscript. This paper has not discussed any of the studies assessing nickel nanoparticles against MRSA. Therefore, there is a lack of data in this manuscript with this regard. We assessed studies regarding anti-MRSA effects of nickel nanoparticles which has not evaluated until today and this is the novelty of our manuscript. However, exact mechanisms of action of nickel nanoparticles against MRSA using molecular or genetic studies and in vivo verification is lacking as we evaluated related eligible published papers.

Round 2

Reviewer 2 Report

Accepted in current form.